# Continual Learners are Viable Long-Tailed Recognizers

## Abstract

We propose a series of theorems which demonstrate that using Continual Learning (CL) to sequentially learn the majority and minority class subsets in a highly imbalanced dataset, is an effective solution for Long-Tailed Recognition (LTR). First, we theoretically prove that under the assumption of strong convexity of the loss function, the weights of a learner trained on a long-tailed dataset are bounded to reside within a neighborhood of the weights of the same learner trained strictly on the largest subset of that dataset. As a result, we present a novel perspective that CL methods, which are designed to optimize the weights in a way that the model performs well on multiple sets, are viable solutions for LTR. To validate our proposed perspective, we first verify the predicted upper bound of the neighborhood radius using the MNIST-LT toy dataset. Next, we evaluate the efficacy of several CL strategies on multiple standard LTR benchmarks (CIFAR100-LT, CIFAR10-LT, and ImageNet-LT), and show that standard CL methods achieve strong performance gains compared to baseline models and tailor-made approaches for LTR. Finally, we assess the applicability of CL techniques on real-world data by exploring CL on the naturally imbalanced Caltech256 dataset and demonstrate its superiority over state-of-the-art models. Our work not only unifies LTR and CL but also paves the way for leveraging advances in CL methods to tackle the LTR challenge more effectively.

## 1 Introduction

Data in real-world scenarios often exhibits long-tailed distributions (Buda et al., 2018; Reed, 2001; Zhang et al., 2023; Fu et al., 2022), where the number of samples in some classes (Head set) is significantly larger than in other classes (Tail set). This imbalance can lead to less than optimal performance in deep learning models. This problem is known as Long-Tailed Recognition (LTR), which can be described as training a model on highly imbalanced data and attempting to achieve high accuracy on a balanced test set (Zhang et al., 2023).

Given that the size of the Head set is substantially larger than the Tail set, samples from the Head generally dominate the loss and determine the gradient. Consequently, samples from the Tail are less impactful, leading to strong performance in Head classes but a significant decline in the performance of the Tail classes (Alshammari et al., 2022). Numerous studies have sought to mitigate this issue by balancing training data through over-sampling the Tail classes (Chawla et al., 2002; Estabrooks et al., 2004; Feng et al., 2021). Alternatively, a feature extractor can be trained using the Head set and employed for transfer learning to train the Tail classifier (Liu et al., 2019; Wang et al., 2017; Zhong et al., 2019; Jamal et al., 2020). As another solution, the loss or gradients have been regularized during training (Cao et al., 2019c; Cui et al., 2019b; Tang et al., 2020b). Recently, weight balancing has been proposed as a method for penalizing excessive weight growth during training, thus forcing per-class weight norms to maintain more uniform magnitudes (Alshammari et al., 2022). While these strategies have achieved notable results in LTR, a comprehensive understanding of the underlying behavior of Deep Learning models in LTR setup remains elusive. This lack of theoretical insight constitutes a gap in the field, making it difficult to systematically improve or interpret model behavior.

In this paper, we present and prove a theorem stating that under the precondition of strong convexity of the loss function, the weights obtained by a learner when trained on the entire dataset consisting of

several subsets with substantially different sizes are confined within an upper bound in relation to the weights achieved by the same learner when trained solely on the largest subset (Head set). We derive that this upper bound is proportional to the imbalance factor of the long-tailed dataset and inversely proportional to the strong convexity parameters of the loss functions. As a result, we demonstrate that learning the whole dataset can be broken down into sequential tasks, i.e., learning the Head followed by the Tail. We therefore propose that Continual Learning (CL) methods can be leveraged to update the weights to learn the second task (Tail) without experiencing forgetting of the first task (Head), which often occurs when a model is retrained. Consequently, we take an interesting step towards unifying these two frameworks (LTR and CL). To provide a theoretical basis for using CL to solve LTR issues, we introduce a further theorem proving that applying CL techniques to adapt the learner to the Tail classes yields a lower loss compared to just naively retraining (i.e. fine-tuning) the learner on the Tail. We validate our theory using five datasets, MNIST-LT, CIFAR100-LT, CIFAR10-LT, ImageNet-LT, and Caltech256. First, we use the toy MNIST-LT dataset and show that the actual distance between weight vectors when trained on either the Head or the entire dataset aligns closely with our theoretical predictions. Next, to further assess the efficacy of employing CL in tackling LTR, we apply a range of CL methods on the LTR problem using CIFAR100-LT, CIFAR10-LT, and ImageNet-LT with varying imbalance factors. The results indicate that CL methods are indeed viable long-tailed recognizers capable of competing with state-of-the-art LTR solutions.

Our contributions in this paper can be summarized as follows: (**1**) We propose a theorem that sets an upper bound on the distance between weights obtained when training a learner on different partitions of an imbalanced dataset, under the assumption of strong convexity of the loss function. This bound is inversely proportional to the imbalance factor and proportional to the strong convexity of the loss function. (**2**) Building on this theorem, we introduce a new perspective on using CL solutions for the LTR problem. To support this perspective, we prove the effectiveness of CL in reducing the loss when learning the Head and Tail sets sequentially. (**3**) We substantiate our method through comprehensive experiments that demonstrate the effectiveness of CL techniques in addressing LTR. Our results indicate significant performance gains in long-tailed scenarios when using standard CL approaches.

## 2 RELATED WORK

**Long-Tailed Recognition.** Real-world datasets often exhibit imbalanced distributions, with some classes appearing more frequently than others. Training a model on such imbalanced data can result in poor performance on the rare classes. LTR addresses this issue by enabling models to perform well on both Head and Tail classes (Cao et al., 2019c). LTR approaches can be broadly categorized into three primary groups: data distribution re-balancing, class-balanced losses, and transfer learning from Head to Tail (Kang et al., 2019). Data distribution re-balancing techniques include over-sampling the Tail (Chawla et al., 2002; Han et al., 2005), under-sampling the Head (Drummond et al., 2003), and class-balanced sampling (Shen et al., 2016; Mahajan et al., 2018). Class-balanced loss approaches modify the loss function to treat each sample differently, e.g., including class distribution-based loss (Cao et al., 2019c; Cui et al., 2019b; Huang et al., 2019), focal loss (Lin et al., 2017b), and Bayesian uncertainty (Khan et al., 2019). Additionally, transfer learning techniques leverage features learned from the Head to improve learning on the Tail (Yin et al., 2019; Liu et al., 2019). More recently, the limitations of class re-balancing have been discussed and the Bilateral-Branch Network (BBN) was proposed to improve representation learning (Zhou et al., 2020). The RoutIng Diverse Experts (RIDE) model was introduced to enhance LTR by reducing model variance (Wang et al., 2020). Finally, the assumption that the test set distribution is always uniform was challenged and test-agnostic long-tailed recognition was introduced (Zhang et al., 2022). It has been discussed that self-supervised learning facilitates universal feature learning, improving performance on test sets with unknown distribution. Although numerous prior works have addressed LTR, few provide a mathematical analysis of the training process using imbalanced data (Ye et al., 2021; Francazi et al., 2023). These works demonstrate that the Head is learned more quickly than the Tail, primarily focusing on the training dynamics. In contrast, our theoretical analysis studies the convergence point of training within the LTR framework. As mentioned earlier, some of the LTR solutions fall into the category of sequential learning, where Head and Tail are learned sequentially. Unlike these works, our work delves into the theoretical foundations of why sequential learning is particularly well-suited for LTR, identifying the key factors that influence the success of these methods. By establishing a mathematical framework, we present a novel perspective on

the applicability of sequential learning to LTR, which has not been considered in prior works. We also introduce CL as a comprehensive solution to the LTR problem for the first time, drawing on broad principles rather than specific techniques. This lends support for, and takes a step towards explaining, the good performance exhibited by some recent sequential CL methods.

**Continual Learning.** CL addresses the challenge of adapting a deep learning model to new tasks (e.g., new classes or distributions) while maintaining performance on the previously learned tasks. The main challenge to address by CL methods is the mitigation of catastrophic forgetting, i.e., forgetting the previous tasks as the new tasks are learned. CL methods are typically grouped into three categories: expansion-based, regularization-based, and memory-based approaches. Expansion-based CL methods utilize a distinct subset of parameters for learning each task (Sarwar et al., 2019; Li et al., 2019; Yoon et al., 2020). Regularization-based techniques penalize significant changes in crucial network parameters (relative to previous tasks) by incorporating a regularization term in the loss function (Saha et al., 2020; 2021; Farajtabar et al., 2020; Kirkpatrick et al., 2017; Li & Hoiem, 2017). Memory-based approaches employ a replay memory to store a limited number of samples from previous tasks, which are then used in future training to minimize forgetting (Riemer et al., 2018; Chaudhry et al., 2019; Shim et al., 2021). Few works attempt to solve both problems of CL and LTR simultaneously in the long-tailed class incremental learning setup. First, a novel replay method called Partitioning Reservoir Sampling (PRS) is proposed (Kim et al., 2020). This method dedicates a sufficient amount of memory to tail classes in order to avoid catastrophic forgetting in minority classes. Class incremental learning is also addressed in a more challenging setup where the new tasks are not uniformly distributed (Liu et al., 2022). In this case, the new tasks are LTR, which makes CL more challenging. They considered two setups: Ordered and Shuffled, where the number of samples in each new task is less than in previous tasks, and when the size of classes is completely random, respectively. More recently, gradient surgery has been employed for addressing CL where the gradient from the new task is projected to the orthogonal direction of the previously learned tasks to ensure learning the new task does not impact the previous task (Saha et al., 2020; Saha & Roy, 2023). These methods achieve state-of-the-art performance on CL benchmarks. Note that none of the above works attempts to employ CL as a solution for LTR scenarios.

## 3   METHOD

### 3.1   OVERVIEW

Let us assume an LTR problem and a learner, denoted as $\theta$. Initially, the learner is trained on a highly imbalanced dataset $\mathcal{D}$, as shown in Fig. 1, where $\theta_i$ is the initialized model in the weight space. Owing to the larger number of Head samples in each iteration, they dominate the evolution of the gradients, resulting in a learner that performs significantly better on the Head set than on the Tail set at the end of training. This process leads the parameters to converge to $\theta^*$. To mitigate this issue, we propose to reformulate the LTR problem as a sequential problem consisting of two tasks: learning the Head and Tail classes separately. Given that the learner already demonstrates strong performance on the Head set, it primarily needs to focus on learning the second task (Tail set). We propose a theorem showing that under a strongly convex loss function, $\theta^*$ lies within a bounded neighborhood radius $r$ of the learner's weights $\theta_H^*$ when trained exclusively on the Head set $\mathcal{D}_H$, where $r$ is proportional to the strong convexity of the loss function and inversely proportional to the imbalance factor. $\psi_H$ represents an area within the weight space where the network performs well on the Head set.

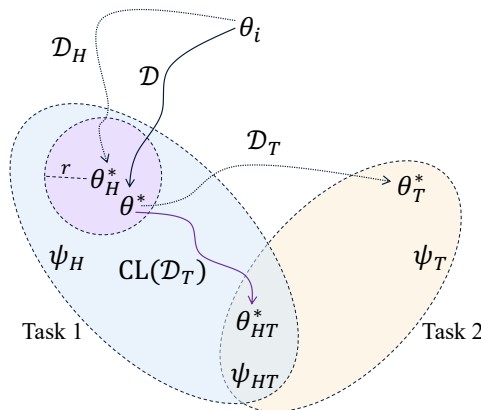

Figure 1: Overview of learning under the LTR scenario and our proposed algorithm (symbols described in text).

However, once the learner attempts to learn these two tasks sequentially, it will encounter another problem known as catastrophic forgetting. Catastrophic forgetting occurs when a deep learning model is trained to perform a new task, but forgets the previous one (Rolnick et al., 2019). Training initially on the Head set followed by training on the Tail set results in $\theta_T^*$, which exhibits catastrophic forgetting. The ideal weights $\theta_{HT}^*$ for learning both Head and Tail sets lie in the intersection of $\psi_H$ and $\psi_T$, denoted by $\psi_{HT}$. To prevent catastrophic forgetting of the first task (Head set) while learning the second task (Tail set), CL techniques can be employed, allowing the model to learn the Tail set without compromising its performance on the Head set. By re-framing LTR as two sequential tasks (learning Head set $\mathcal{D}_H$ followed by Tail set $\mathcal{D}_T$), we can utilize CL to learn the second task (updating the weights towards $\psi_T$ with $\mathrm{CL}(\mathcal{D}_T)$ without forgetting the first task (staying in $\psi_H$), ultimately performing well on both Head and Tail sets (ending up in $\psi_{HT}$).

## 3.2 PROBLEM FORMULATION

LTR aims to address the challenge of learning from highly imbalanced data. This occurs when the training data $\mathcal{D}$ contains more samples in some classes (the Head set) and fewer in others (the Tail set). Let $\mathcal{D}_H$ and $\mathcal{D}_T$ represent the subsets of $\mathcal{D}$ corresponding to the Head set and Tail set, respectively. The imbalance factor *IF* quantifies the severity of this issue in a dataset:

$$IF = \frac{|\mathcal{D}_{c^{\max}}|}{|\mathcal{D}_{c^{\min}}|}, \tag{1}$$

where $c$ represents the class index, $|\mathcal{D}_c|$ denotes the cardinality of each class, $c^{\max} = \arg\max |\mathcal{D}_c|$, and $c^{\min} = \arg\min |\mathcal{D}_c|$, such that $\mathcal{D}_{c^{\max}} \in \mathcal{D}_H$ and $\mathcal{D}_{c^{\min}} \in \mathcal{D}_T$.

**Definition 3.1.** *A dataset is deemed* long-tailed *when* $|\mathcal{D}_{c^{\max}}| \gg |\mathcal{D}_{c^{\min}}|$ *or, in other words, IF $\gg$ 1. When a model is trained on such a dataset and its performance is assessed on a uniformly distributed test set (i.e. $|\mathcal{D}_c| = k$ for each class $\mathcal{D}_c$ within the test set), the problem is referred to as* Long-Tailed Recognition.

## 3.3 TRAINING ON LONG-TAILED DISTRIBUTION

In this section, we derive the conditions in which CL can be applied to a long-tailed scenario. We assume that all head classes are of size $|\mathcal{D}_H|$, and all tail classes are of size $|\mathcal{D}_T|$, with $|\mathcal{D}_H| >> |\mathcal{D}_T|$. The training process in the LTR setup is analyzed using the following Theorem.

**Theorem 3.2.** *Assume that a logistic regression model with parameters $\theta$ is trained using regularized cross-entropy loss in an LTR setting. Then, $\|\theta^* - \theta_H^*\|^2 \le \frac{4\delta}{\mu_H + \mu}$, where $\theta^*$ represents the parameter vector obtained after training, $\theta_H^*$ denotes the parameter vector when the model is trained solely on the Head set, $\delta$ is the maximum difference between the loss of the learner using the entire dataset or the Head set for any value of $\theta$, and $\mu_H$ and $\mu$ are the strong convexity parameters of the loss computed on either the Head set or the entire dataset.*

*Proof.* The model is trained on the entire dataset $\mathcal{D}$ by minimizing the loss function $\mathcal{L}$:

$$\mathcal{L}(\mathcal{D}) = \frac{1}{|\mathcal{D}|} \left( \sum_{i=1}^{|\mathcal{D}_H|} \ell(\mathcal{D}_H^i) + \sum_{i=1}^{|\mathcal{D}_T|} \ell(\mathcal{D}_T^i) \right), \tag{2}$$

where $\ell(\mathcal{D}^i)$ is the loss of each individual sample. By substituting $\mathcal{L}(\mathcal{D}_H) = \frac{1}{|\mathcal{D}H|} \sum_{i=1}^{|\mathcal{D}_H|} \ell(\mathcal{D}_H^i)$ and $\mathcal{L}(\mathcal{D}_T) = \frac{1}{|\mathcal{D}T|} \sum_{i=1}^{|\mathcal{D}_T|} \ell(\mathcal{D}_T^i)$ :

$$\mathcal{L}(\mathcal{D}) = \frac{|\mathcal{D}_H|}{|\mathcal{D}|} \mathcal{L}(\mathcal{D}_H) + \frac{|\mathcal{D}_T|}{|\mathcal{D}|} \mathcal{L}(\mathcal{D}_T). \tag{3}$$

We define $\gamma = \frac{IF}{1+IF}$, which falls within the range of $[0.5, 1)$. We can rewrite Eq. 3 as:

$$\mathcal{L}(\mathcal{D}) = \gamma \mathcal{L}(\mathcal{D}_H) + (1 - \gamma)\mathcal{L}(\mathcal{D}_T). \tag{4}$$

Since *IF* $\gg 0$ in LTR, we can conclude that the value of $\gamma$ approaches one. Consequently, $\mathcal{L}(\mathcal{D})$ approaches $\mathcal{L}(\mathcal{D}_H)$ for all $\theta$ values. Let $\delta$ be defined as the maximum difference of the losses:

$$|\mathcal{L}(\mathcal{D}) - \mathcal{L}(\mathcal{D}_H)| \le \delta. \tag{5}$$

From Eq. 4, it follows that $\lim_{IF \gg 0} \delta = 0$.

One of the most effective losses for the LTR problem is the regularized cross-entropy loss. This loss is the cross-entropy with an additional regularization term that prevents weights from growing excessively:

$$\mathcal{L}(\mathcal{D}, \theta) = -\frac{1}{N} \sum_{i=1}^{N} y_i \log \left( P(f(\theta, x_i)) \right) + \frac{\mu}{2} \|\theta\|^2, (x_i, y_i) \in \mathcal{D}. \tag{6}$$

This loss improves generalizability by reducing overfitting and achieves state-of-the-art performance when dealing with LTR scenarios (Alshammari et al., 2022). Moreover, as our model is logistic regression, this loss is strongly convex since $\nabla^2 \mathcal{L}(\beta, \theta) \geq \mu$. From the definition of strong convexity (Sherman et al., 2021), it therefore follows that:

$$\mathcal{L}(x_1) \geq \mathcal{L}(x_2) + \nabla \mathcal{L}(x_2)^T (x_1 - x_2) + \frac{\mu_{\mathcal{L}}}{2} \|x_1 - x_2\|^2, \tag{7}$$

where $\mu_{\mathcal{L}}$ is the strong convexity parameter. Now, we are introducing Lemma 3.3:

**Lemma 3.3.** *If $|f(x) - g(x)| \leq \delta$ and both $f(x)$ and $g(x)$ are strongly convex then:*

$$\|x_g - x_f\|^2 \leq \frac{4\delta}{\mu_f + \mu_g}, \tag{8}$$

*where $x_g$ and $x_f$ are $\arg\min f(x)$ and $\arg\min g(x)$, respectively. The proof of this lemma is presented in Appendix A.1.*

Applying Lemma 3.3 to Eqs. 5 and 7 yields:

$$\|\theta^* - \theta_H^*\|^2 \leq \frac{4\delta}{\mu_H + \mu}, \tag{9}$$

where $\theta^*$ and $\theta_H^*$ are $\arg\min \mathcal{L}$ and $\arg\min \mathcal{L}_H$, respectively. $\qquad \square$

As a result, when the model is trained on a long-tailed dataset, the network parameter $\theta$ converges to a point close to the weights of the model when it was only trained on the Head set $\theta_H$.

**Remark 3.4.** *Under a more relaxed assumption, where $\mathcal{L}(\mathcal{D}, \theta)$ is strictly (but not strongly) convex, the upper bound can be calculated using Lemma 3.5.*

**Lemma 3.5.** *If $|f(x) - g(x)| \leq \delta$ and both $f(x)$ and $g(x)$ are strictly convex then:*

$$\|x_g - x_f\|^2 \leq \frac{4\delta}{\lambda_f + \lambda_g}, \tag{10}$$

*where $x_g$ and $x_f$ are $\arg\min f(x)$ and $\arg\min g(x)$, and $\lambda_f$ and $\lambda_g$ are the minimum eigenvalues of the hessian matrices of $f(x)$ and $g(x)$, respectively. The full proof is provided in Appendix A.5.*

Using lemma 3.5, the upper bound of $\|\theta^* - \theta_H^*\|^2$ is expressed as $\frac{4\delta}{\lambda_f + \lambda_g}$. To ensure that this upper bound is limited and approaches zero when $\delta \to 0$, the minimum eigenvalues of the Hessians of both loss functions should have lower bounds, which is again another definition of strong convexity.

Theorem 3.2 assumes that there is only one Head and one Tail in the dataset, which is not the case in many real-world datasets. In cases where we relax the assumption to allow the Head and Tail sets to follow a long-tailed distribution, these sets can be further partitioned into distinct Head and Tail subsets. While each individual partition remains imbalanced, we continue to subdivide them until $IF_{D^i} \not\gg 1$ for all partitions $D^i$. In this scenario, there is no long-tailed partition of the data. Theorem 3.6 extends Theorem 3.2 to address this scenario for any number of partitions.

**Theorem 3.6.** *Let a logistic regression model with parameters $\theta$ be trained using regularized cross-entropy loss in an LTR setting, and let dataset $\mathcal{D}$ be divided into $n$ partitions. Further, let a subset of $m < n$ partitions be $\bigcup_{i=1}^{m} \mathcal{D}_i \subseteq \mathcal{D}$, with the largest partition being $\mathcal{D}_a$, i.e. $a = \arg\max_i |\mathcal{D}_i|, i \in [1, m]$. Then, the weights $\theta_{\bigcup \mathcal{D}_i}^*$ obtained from training the model on $\bigcup_{i=1}^{m} \mathcal{D}_i$ will always be in a bounded neighborhood of the weights $\theta_{\mathcal{D}_a}^*$ obtained from training on the largest subset $\mathcal{D}_a$.*

*Proof Sketch.* (Formal proof in Appendix A.3) We start by dividing the dataset into multiple partitions and focus on the two largest subsets. Using Theorem 3.2, we establish a bound on the weight differences when training on these two subsets. We then consider the aggregation of these two subsets as a new 'largest' subset and compare it with the next largest partition, iteratively applying Theorem 3.2. This allows us to calculate an upper bound on the weight differences when training on all subsets versus only the largest one.

## 3.4 CL FOR LTR

A general CL problem can be formulated as follows (Prabhu et al., 2020). A model is exposed to streams of training samples $(x_t, y_t)$, where $t$ represents the time step. The set of data labels $\mathcal{Y}_t = \bigcup_{i=1}^{t} y_i$ has been seen by the network previously, up to the current timestep $t$. The objective at any timestep is to find a mapping $f_\theta : x \rightarrow y$ that accurately maps sample $x$ to $\mathcal{Y}_t \cup y_{t+1}$, where $y_{t+1}$ is the set of new unseen labels.

We have shown in Eq. 9 that when the model is trained on highly imbalanced data, the weights $\theta^*$ will be very close to those weights $\theta_H^*$ when it is only trained on the Head set. As a result, the model can be considered as $f_{\theta_H^*} : x \rightarrow y$ where $\mathcal{Y}_t = \mathcal{D}_H$. The objective is to learn $f_\theta : x \rightarrow y$ which could accurately predict the entire dataset $\mathcal{D}$. Thus, if we consider the Tail set $\mathcal{D}_T$ as $y_{t+1}$, the objective of the LTR problem would be equivalent to the objective of CL, which is to estimate $f_{\theta_t}$:

$$f_{\theta_t} : x \rightarrow y \quad s.t. \quad y \in \mathcal{Y}_t \cup y_{t+1}, \ \mathcal{Y}_t = \mathcal{D}_H, \ y_{t+1} = \mathcal{D}_T. \tag{11}$$

This approach unifies the two domains so that an LTR problem can be treated as a CL problem. In order to prove the effectiveness of employing CL methods for addressing LTR problems, the following theorem is proposed.

**Theorem 3.7.** *Consider a logistic regression model with parameters $\theta$ trained using regularized cross-entropy loss in an LTR setting, converging to $\theta^i$. Then, $\mathcal{L}(\mathcal{D}, \theta_{EWC}^{i+1}) < \mathcal{L}(\mathcal{D}, \theta_{\mathcal{L}}^{i+1})$, where $\theta_{EWC}^{i+1}$ and $\theta_{\mathcal{L}}^{i+1}$ denote the weights of the model after a single update using EWC loss and regularized cross-entropy loss, respectively.*

*Proof Sketch.* (Formal proof in Appendix A.4) We consider the updated weights after one iteration using both EWC loss and regularized cross-entropy loss. By employing Taylor expansion, we approximate the losses for the new weights. We then show that the EWC loss incorporates a regularization term that effectively constrains the weight updates. Leveraging the strong convexity of the loss function and the positive nature of the Fisher information matrix, we prove that the loss with EWC-updated weights is strictly less than that with regular cross-entropy updated weights.

This theorem demonstrates that even a fundamental CL method like EWC can enhance the performance of a logistic regression model trained with regularized cross-entropy loss. It's important to highlight that while EWC is not the most recent CL technique to be proposed, it serves as a common baseline for comparison of all other CL methods. Furthermore, the mathematical formulation of EWC is succinct, and is amenable for use within Theorem 3.7. As this fundamental CL method proves effective at tackling LTR challenges, more recent CL approaches are likely to be even more beneficial, as we will show in the experimental results in the following section.

## 4 EXPERIMENTS AND RESULTS

### 4.1 EXPERIMENT SETUP

**Datasets.** First, we use the **MNIST-LT** (LeCun et al., 1998) toy dataset with different *IF* values and strong convexity parameters to study the behavior of the upper bound and compliance with our theorem. Next, to evaluate the performance of CL in addressing LTR, we employ three widely used LTR datasets: **CIFAR100-LT**, **CIFAR10-LT** (Cao et al., 2019c), and **ImageNet-LT** (Liu et al., 2019). These datasets represent long-tailed versions of the original CIFAR100, CIFAR10, and ImageNet datasets, maintaining the same number of classes while decreasing the number of samples per class using an exponential function. Finally, to highlight the benefits of using CL for LTR, we carry out additional experiments using the naturally skewed **Caltech256** dataset (Griffin et al., 2007).

**Implementation Details.** Following the experimental setup of (Alshammari et al., 2022) and (Fu et al., 2022), we use ResNet-32 (He et al., 2016) and ResNeXt-50 (Xie et al., 2017) for CIFAR and

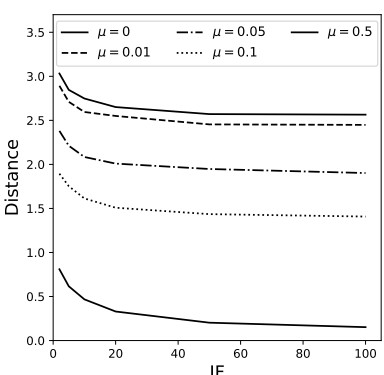

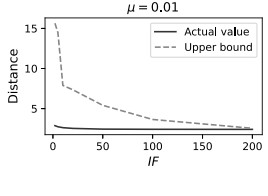
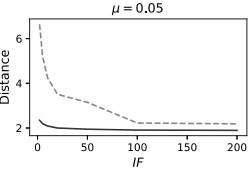
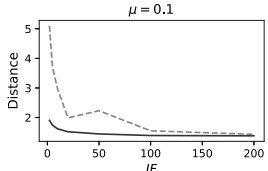
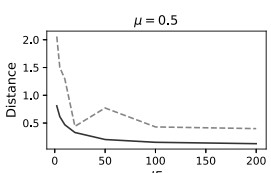

Figure 2: The distance between $\theta^*$ and $\theta_H^*$ in different *IF* and $\mu$.

Figure 3: The actual distance between $\theta^*$ and $\theta_H^*$ in different *IF* and $\mu$ compared with the calculated upper bound.

ImageNet benchmarks, respectively. The LTR methods selected for comparison are state-of-the-art solutions in the area. All training was conducted using an NVIDIA RTX 3090 GPU with 24GB VRAM. The details of the implementation specifics are provided in Appendix B.

**Evaluation.** For the LTR datasets (MNIST-LT, CIFAR100-LT, CIFAR10-LT, ImageNet-LT), we first train the model on the long-tailed imbalanced training set and then evaluate it on the balanced test set, following the evaluation protocol of (Alshammari et al., 2022). For Caltech256, we use the entire training set for training and assess the model's performance on the entire test set, retaining its original distribution. All reported values represent classification accuracy percentages.

## 4.2 RESULTS

**Upper bound.** To investigate the distance between the acquired sets of weights by training on $\mathcal{D}$ and $\mathcal{D}_H$ measured by $\|\theta^* - \theta_H^*\|$, we first train a logistic regression model on MNIST-LT with varying *IF* and $\mu$ values. Then we calculate the Euclidean distance between the two sets of weights, as illustrated in Fig. 2. As expected from Eq. 9, increasing either the *IF* or strong convexity ($\mu$) results in a reduced distance, indicating that the weights of the model trained using $\mathcal{D}$ approach the weights when it is solely trained using $\mathcal{D}_H$.

To verify the upper bound in Eq. 9, we then calculate the estimated upper bound for each $\gamma$ and $\mu$ using Eq. 5 in Appendix A.1. It is important to note that this upper bound is tighter compared to Eq. 9 . We compare the upper bound with the actual distance in Fig. 3 and show that for all *IF* and $\mu$ values, the measured distance is lower than the theoretical upper bound.

**LTR benchmarks.** To demonstrate the efficacy of CL approaches in addressing the LTR challenge, we apply five commonly used CL strategies (LwF (Li & Hoiem, 2017), EWC (Kirkpatrick et al., 2017), Modified EWC (Molahasani et al., 2023), GPM (Saha et al., 2020), and SGP (Saha & Roy, 2023)) on three LTR benchmark datasets (CIFAR100-LT, CIFAR10-LT, and ImageNet-LT). The number of samples in each class decreases exponentially according to the *IF*, where first class has the maximum number of samples and the last class contains the least number of samples, as illustrated in Appendix C. The results are presented in Table 1, Table 2, Table 3 along with the performance of existing LTR solutions, specifically designed and implemented for this problem. Moreover, we present two baselines by training the ResNet32 and ResNeXt-50 encoder on the imbalanced data, with and without a class-balanced loss term. The accuracies presented in the tables represent the average per-class accuracies. We observe that CL methods indeed provide an effective solution for LTR, as predicted by our proposed theorems. We acknowledge that the CL approaches may not yield top-performing solutions in contrast to certain existing LTR methods. However, when compared to the baselines, the CL methods still demonstrate a considerable improvement in performance. The superior performance of some of the LTR methods can be credited to their tailored design for this particular benchmark, along with the likelihood that the strong convexity assumption may not hold perfectly for this experiment. Some LTR solutions like BBN (Zhou et al., 2020) and SADE

Table 1: LTR benchmarks for CIFAR100-LT.

| Model | IF 100 | 50 | 10 |
|---|---|---|---|
| Baselines | | | |
| Cui et al. (2019a)[1] | 38.3 | 43.9 | 55.7 |
| Cui et al. (2019a)[2] | 39.6 | 45.3 | 58.0 |
| LTR methods | | | |
| Cui et al. (2019a)[3] | 39.6 | 45.2 | 58.0 |
| Kang et al. (2019) | 47.7 | 52.5 | 63.8 |
| Cao et al. (2019a) | 42.0 | 46.6 | 58.7 |
| Zhou et al. (2020) [*] | 42.6 | 47.0 | 59.1 |
| Menon et al. (2020) | 42.0 | 47.0 | 57.7 |
| Yang & Xu (2020) | 43.4 | 47.1 | 58.9 |
| Tang et al. (2020a) | 44.1 | 50.3 | 59.6 |
| Li et al. (2021a) | 46.0 | 50.5 | 62.3 |
| He et al. (2021) | 45.4 | 51.1 | 62.0 |
| Samuel & Chechik (2021) | 47.3 | 57.6 | 63.4 |
| Cui et al. (2021) | 52.0 | 56.0 | 64.2 |
| Zhang et al. (2022) [*] | 49.8 | 53.9 | 63.9 |
| Alshammari et al. (2022)[4] | 46.0 | 52.7 | 66.0 |
| Alshammari et al. (2022)[5] | 53.4 | 57.7 | 68.7 |
| CL methods | | | |
| Li & Hoiem (2017) | 45.1 | 49.3 | 58.7 |
| Kirkpatrick et al. (2017) | 44.4 | 50.3 | 58.8 |
| Molahasani et al. (2023) | 45.9 | 51.0 | 60.7 |
| Saha et al. (2020) | 47.9 | 53.2 | 63.3 |
| Saha & Roy (2023) | 50.0 | 55.9 | 66.1 |

Table 2: LTR benchmarks for CIFAR10-LT.

| Model | IF 100 | 50 |
|---|---|---|
| Baselines | | |
| Cui et al. (2019a)[1] | 69.8 | 75.2 |
| Cui et al. (2019a)[2] | 74.7 | 79.3 |
| LTR methods | | |
| Lin et al. (2017a) | 70.4 | 75.3 |
| Zhang et al. (2018) | 73.1 | 77.8 |
| Cui et al. (2018) | 67.1 | 75.0 |
| Díaz-Rodríguez et al. (2018) | 85.2 | 88.2 |
| Cui et al. (2019a) | 74.6 | 79.3 |
| Cao et al. (2019a) | 77.0 | 79.3 |
| Cao et al. (2019b) | 79.8 | 82.2 |
| Cui et al. (2019a)[6] | 73.0 | 78.1 |
| Cui et al. (2019a)[7] | 78.1 | 82.4 |
| Zhou et al. (2020) [*] | 79.8 | 82.2 |
| Tang et al. (2020a) | 80.6 | 83.6 |
| Wang et al. (2021b) | 81.4 | 85.4 |
| Zhong et al. (2021) | 82.1 | 85.7 |
| Zhu et al. (2022) | 84.3 | 87.2 |
| CL methods | | |
| Li & Hoiem (2017) | 76.3 | 78.6 |
| Kirkpatrick et al. (2017) | 75.1 | 80.1 |
| Molahasani et al. (2023) | 77.8 | 81.3 |
| Saha et al. (2020) | 81.2 | 84.8 |
| Saha & Roy (2023) | 83.0 | 85.5 |

Table 3: LTR benchmarks for ImageNet-LT.

| Model | Top-1 Accuracy |
|---|---|
| Baselines | |
| Cui et al. (2019a)[1] | 44.4 |
| Cui et al. (2019a)[2] | 33.2 |
| LTR methods | |
| Hinton et al. (2015) | 35.8 |
| Lin et al. (2017a) | 30.5 |
| Mahajan et al. (2018) | 46.8 |
| Liu et al. (2018) | 35.6 |
| Kang et al. (2019)[8] | 49.6 |
| Kang et al. (2019)[9] | 49.4 |
| Xiang et al. (2020) | 37.5 |
| Tang et al. (2020a) | 51.8 |
| Wang et al. (2021a) | 50.4 |
| He et al. (2021) | 53.1 |
| Samuel & Chechik (2021) | 53.5 |
| Zhang et al. (2021) | 52.9 |
| Alshammari et al. (2022)[4] | 48.6 |
| Alshammari et al. (2022)[5] | 53.9 |
| CL methods | |
| Li & Hoiem (2017) | 47.6 |
| Kirkpatrick et al. (2017) | 48.9 |
| Molahasani et al. (2023) | 49.1 |
| Saha et al. (2020) | 50.6 |
| Saha & Roy (2023) | 52.0 |

[1]Baseline, [2]Baseline + CB, [3]Focal+CB, [4]WD, [5]WD + MAX, [6]Manifold mixup, [7]ELF (LDAM)+DRW, [8]cRT, [9]$\tau$-norm, [*] sequential methods.

(Zhang et al., 2022) learn the Head and Tail sequentially, and rely on various techniques to prevent performance loss on the Head while learning the Tail. However, our results in Tables 1 to 2 (LTR sequential solutions are highlighted by [*].) demonstrate that current CL methods designed to mitigate catastrophic forgetting while being trained sequentially are competitive solutions for LTR. Note that prior works such as (Alshammari et al., 2022) avoid direct comparisons with solutions such as RIDE (Wang et al., 2020), ACE (Cai et al., 2021), SSD (Li et al., 2021b), and PaCo (Cui et al., 2021) that employ data augmentations, ensembles, and self-supervised pretraining, as these techniques can also be applied to other solutions to gain performance boosts. Hence, such methods are not included in our tables.

**Additional discussions.** Prior works discuss three key concepts in the context of CL: catastrophic forgetting, backward transfer, and forward transfer (Díaz-Rodríguez et al., 2018). As mentioned earlier, catastrophic forgetting occurs when the performance of a class declines after retraining. Despite the use of CL methods, which are designed to mitigate this forgetting, a certain degree of forgetting is still inevitable. Forward transfer is the improvement in performance on a new task after employing CL, which is the central aim of retraining in CL. Finally, backward transfer is a beneficial side-effect where retraining on new samples can actually enhance the model's performance on the previous tasks. Now, let's discuss Fig. 4, which presents the difference in per-class accuracy of the best CL method (SGP) versus the baseline network. The analysis is based on CIFAR100-LT with an *IF* of 100. The figure is divided into three regions corresponding to the scenarios discussed above: catastrophic forgetting (bottom), backward transfer (top-left), and forward transfer (top-right). The bottom region in the figure represents classes that undergo catastrophic forgetting, while the top-right region represents the Tail samples (with a class index larger than 60), which demonstrate improved performance, or forward transfer. We observe that using SGP as a CL solution for LTR results in

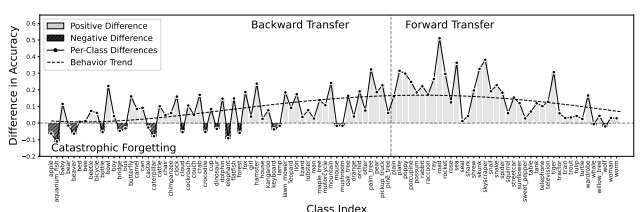

Figure 4: The difference in per-class accuracy of SGP and the baseline model. 🔍

very effective improvements in the per-class accuracy of the Tail (forward transfer). Interestingly, despite the absence of Head data in the retraining process, 42 out of 60 Head classes see some level of improvement after the model is exposed to the Tail samples (backward transfer). This result emphasizes the remarkable potential of CL methods in enhancing the performance on both new and previous tasks.

The inference runtime is identical between CL-based methods and LTR solutions, due to identical backbones in both types of methods and the fact that CL does not affect inference. Regarding the training runtime, when CL is used to address LTR, the data is divided into Head and Tail sets. At each step of the training, only one partition of data is involved. Since the backbone is consistent among all LTR approaches for each benchmark, the runtime is determined by the amount of data fed to the model. Dividing the learning into multiple steps and using CL therefore does not impact the total runtime, nor does it increase the training time.

**Real-world data.** In LTR benchmarks, datasets are modified to exhibit a skewed distribution of samples among various classes. However, such imbalanced class distributions are naturally observed in real-world data as well. To evaluate the efficacy of CL techniques on non-LTR benchmark datasets, we utilize the Caltech256 dataset (Griffin et al., 2007), which consists of 256 distinct classes representing everyday objects. The largest class comprises 827 samples, while the smallest class contains only 80 samples, exhibiting an *IF* of over 10. Here, we employ the CL solution, Modified EWC, and compare its performance to state-of-the-art methods on this dataset for objected classification. The results are presented in Table 4. We observe that CL outperforms the state-of-

Table 4: The performance of CL compared with SOTA models on Caltech256.

| Method | Backbone | |
|---|---|---|
| | Inc.V4 | Res.101 |
| $L^2 - FE$ (Li et al., 2018) | 84.1% | 85.3% |
| $L^2$ (Li et al., 2018) | 85.8% | 87.2% |
| $L^2 - SP$ (Li et al., 2018) | 85.3% | 87.2% |
| DELTA (Li et al., 2018) | 86.8% | 88.7% |
| TransTailor (Liu et al., 2021) | - | 87.3% |
| Continual Learning | 87.56% | 88.9% |

the-art on this dataset, demonstrating the strong potential of using CL in dealing with long-tailed real-world datasets.

**Limitations.** Strong convexity is a key assumption in our theorem, which determines an upper bound for the distance between the weights of a learner trained on the full dataset and the weights of the same learner trained solely on the Head. This assumption offers a solid theoretical foundation for our method, showcasing the feasibility of using CL techniques to address the LTR problem. However, as many deep learning models in practice employ non-convex loss functions that potentially limit the theorem's applicability to specific cases, it is crucial to highlight that our experimental results are not strictly dependent on the strong convexity condition. In fact, our method exhibits impressive performance even under more relaxed conditions, indicating its robustness and adaptability.

## 5 CONCLUSION AND FUTURE WORK

In this work, we address the under-explored behavior of deep learning models in LTR setups. To this end, we proved a theorem establishing an upper bound for the distance between the weights of a learner training on the entire dataset and those trained only on the largest subset. Based on these findings, we then propose a novel perspective of employing CL for addressing LTR and proving its effectiveness. Our experimental results on benchmark datasets like MNIST-LT, CIFAR100-LT, CIFAR10-LT, and ImageNet-LTR verify our theoretical findings and demonstrate the viability of our approach in achieving competitive performances as compared to baselines and state-of-the-art LTR solutions. We also showcase the applicability of CL techniques to real-world data by employing CL on the naturally imbalanced Caltech256 dataset, highlighting its performance advantage compared to existing methods. Future research directions include exploring non-convex loss functions to relax our strong convexity assumption and leveraging our findings to design a novel CL-based method specifically engineered for the LTR problem. Ultimately, our findings could help researchers design more robust and scalable solutions for learning from highly imbalanced data, enabling more accurate predictions and generalizations across a wide range of applications.

## REPRODUCIBILITY STATEMENT

To ensure the reproducibility of our work, several key steps have been taken. First, all assumptions and complete proofs of theorems and lemmas are provided in detail in the manuscript and the appendices. Second, for all the algorithms employed in this study, we refer readers to Appendix B where links to comprehensive code repositories are provided. Third, we report all the hyperparameters for each algorithm in full detail in Appendix B, to facilitate replication of our experimental results. Fourth, we have carefully cited and referred to related studies to provide a comprehensive view of the existing literature. Lastly, our experimental setup is designed to closely follow the methodologies employed in related works, ensuring ease of replication and comparability of results. These measures collectively contribute to the robustness and reproducibility of our findings.

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

APPENDIX

## A    PROOFS

### A.1    PROOF OF LEMMA 3.3

*Proof.* Since $f(x)$ is strongly convex:

$$f(x_2) \geq f(x_1) + \nabla f(x_1)^T(x_2 - x_1) + \frac{\mu_f}{2}\|x_2 - x_1\|^2. \tag{1}$$

Accordingly if $x_2 = x_g = \arg\min g(x)$ and $x_1 = x_f = \arg\min f(x)$, then:

$$f(x_g) - f(x_f) \geq \nabla f(x_f)^T(x_g - x_f) + \frac{\mu_f}{2}\|x_g - x_f\|^2. \tag{2}$$

Since $x_f$ is the minimizer of $f$, $\nabla f(x_f) = 0$. Therefore:

$$f(x_g) - f(x_f) \geq \frac{\mu_f}{2}\|x_g - x_f\|^2. \tag{3}$$

Similarly, considering $g(x)$, with $x_1 = x_g$, and $x_2 = x_f$, we can derive Equation 1 as follows:

$$g(x_f) - g(x_g) \geq \frac{\mu_g}{2}\|x_f - x_g\|^2. \tag{4}$$

By adding and rearranging Eqs. 3 and 4, we will have:

$$(g(x_f) - f(x_f)) + (f(x_g) - g(x_g)) \geq \frac{(\mu_f + \mu_g)}{2}\|x_g - x_f\|^2. \tag{5}$$

Using $|f(x) - g(x)| \leq \delta$, we can maximize $(g(x_f) - f(x_f))$ and $(f(x_g) - g(x_g))$ to obtain:

$$2\delta \geq \frac{\mu_f + \mu_g}{2}\|x_g - x_f\|^2. \tag{6}$$

Hence:

$$\|x_g - x_f\|^2 \leq \frac{4\delta}{\mu_f + \mu_g}, \tag{7}$$

which completes the proof.    □

### A.2    PROOF OF LEMMA 3.5

*Proof.* Using the second-order Taylor series expansion for multivariate functions, we can approximate $f(x_g)$ and $g(x_f)$ as follows:

$$f(x_g) \simeq f(x_f) + \nabla f(x_f)(x_g - x_f) + \frac{1}{2}(x_g - x_f)^\top H_f(x_f)(x_g - x_f), \tag{8}$$

$$g(x_f) \simeq g(x_g) + \nabla g(x_g)(x_f - x_g) + \frac{1}{2}(x_f - x_g)^\top H_g(x_g)(x_f - x_g), \tag{9}$$

where $H_f(x_f)$ and $H_g(x_g)$ are the Hessian matrices of $f$ and $g$ evaluated at $x_f$ and $x_g$, respectively. Since $\nabla f(x_f) = \nabla g(x_g) = 0$, by adding Eq. 8 and Eq. 9 together, we obtain:

$$f(x_g) - g(x_g) + g(x_f) - f(x_f) \simeq \frac{1}{2}(x_g - x_f)^\top H_f(x_f)(x_g - x_f) + \frac{1}{2}(x_f - x_g)^\top H_g(x_g)(x_f - x_g), \tag{10}$$

Using $|f(x) - g(x)| \leq \delta$, we can maximize $(g(x_f) - f(x_f))$ and $(f(x_g) - g(x_g))$:

$$2\delta \geq \frac{1}{2}(x_g - x_f)^\top H_f(x_f)(x_g - x_f) + \frac{1}{2}(x_f - x_g)^\top H_g(x_g)(x_f - x_g), \tag{11}$$

Let $\lambda_f$ and $\lambda_g$ be the minimum eigenvalues of $H_f(x_f)$ and $H_g(x_g)$, respectively. By properties of the minimum eigenvalues, we can say:

$$(x_g - x_f)^\top H_f(x_f)(x_g - x_f) \geq \lambda_f\|x_g - x_f\|^2, \tag{12}$$

$$(x_f - x_g)^\top H_g(x_g)(x_f - x_g) \geq \lambda_g \|x_f - x_g\|^2. \tag{13}$$

Using Eqs. 12 and 13, we can rewrite Eq. 11:

$$2\delta \geq \frac{1}{2}\lambda_f \|x_g - x_f\|^2 + \frac{1}{2}\lambda_g \|x_f - x_g\|^2. \tag{14}$$

Therefore:

$$\|x_f - x_g\|^2 \leq \frac{4\delta}{\lambda_f + \lambda_g}, \tag{15}$$

which completes the proof. □

### A.3 PROOF OF THEOREM 3.6

*Proof.* Let $\mathcal{D}$ be a dataset divided into a sequence of partitions $\mathcal{D}_1, \mathcal{D}_2, \ldots, \mathcal{D}_n$ such that the imbalance factor between any two consecutive partitions $\mathcal{D}_i$ and $\mathcal{D}_{i+1}$ is significantly large, i.e., $\frac{|\mathcal{D}_i|}{|\mathcal{D}_{i+1}|} \gg 1$.

Consider a random subset of $\mathcal{D}$ sorted from largest to smallest denoted as $\mathcal{D}_a, \mathcal{D}_b, \mathcal{D}_c, \ldots$ (where $|\mathcal{D}_a| \gg |\mathcal{D}_b| \gg |\mathcal{D}_c|$).

From Theorem 3.2, we know that if the imbalance factor between two partitions is significantly large, $\frac{|\mathcal{D}_1|}{|\mathcal{D}_2|} \gg 1$, then the distance between the optimal parameters when trained on $\mathcal{D}_1$ and $\mathcal{D}_1 \cup \mathcal{D}_2$ is bounded by $\zeta$, i.e., $\|\theta^*_{\mathcal{D}_1} - \theta^*_{\mathcal{D}1 \cup \mathcal{D}2}\|^2 \leq \zeta$ where $\zeta$ is computed using Eq. 9 in the manuscript.

Applying this Theorem to $\mathcal{D}_a$ and $\mathcal{D}_b$, we have:

$$\|\theta^*_{\mathcal{D}_a} - \theta^*_{\mathcal{D}_a \cup \mathcal{D}_b}\|^2 \leq \zeta_1$$

Next, considering the combination of $\mathcal{D}_a \cup \mathcal{D}_b$ and $\mathcal{D}_c$, given that $\frac{|\mathcal{D}_a \cup \mathcal{D}_b|}{|\mathcal{D}_c|} \gg 1$, we deduce:

$$\|\theta^*_{\mathcal{D}_a \cup \mathcal{D}_b} - \theta^*_{\mathcal{D}_a \cup \mathcal{D}_b \cup \mathcal{D}_c}\|^2 \leq \zeta_2$$

Given that the weights reside in a metric space, and the distances are Euclidean, the triangle inequality applies. Combining the above inequalities, we therefore get:

$$\|\theta^*_{\mathcal{D}_a} - \theta^*_{\mathcal{D}_a \cup \mathcal{D}_b \cup \mathcal{D}_c}\|^2 \leq (\sqrt{\zeta_1} + \sqrt{\zeta_2})^2$$

Extending this argument for all partitions, we can conclude:

$$\|\theta^*_{\mathcal{D}_a} - \theta^*_{\sum \mathcal{D}_i}\|^2 \leq (\sum_{i=1}^{m} \sqrt{\zeta_i})^2$$

where $m$ is the number of subsets selected randomly. □

### A.4 PROOF OF THEOREM 3.7

*Proof.* Define the updated weight vector after one iteration over the Tail using EWC loss as:

$$\theta^{i+1}_{\text{EWC}} = \theta^i - \eta \nabla \mathcal{L}_{\text{EWC}}(\mathcal{D}_T, \theta^i) \tag{16}$$

Similarly, for $\mathcal{L}$:

$$\theta^{i+1}_{\mathcal{L}} = \theta^i - \eta \nabla \mathcal{L}(\mathcal{D}_T, \theta^i) \tag{17}$$

From the Taylor series expansion, we can estimate the $\mathcal{L}$ of the model with $\theta^{i+1}_{\text{EWC}}$ over $\mathcal{D}$:

$$\mathcal{L}(\mathcal{D}, \theta^{i+1}_{\text{EWC}}) \simeq \mathcal{L}(\mathcal{D}, \theta^i) - \eta \nabla \mathcal{L}_{\text{EWC}}(\mathcal{D}_T, \theta^i) \nabla \mathcal{L}(\mathcal{D}, \theta^i) \tag{18}$$

Similarly, for the $\mathcal{L}$ of the model with $\theta^{i+1}_{\mathcal{L}}$ over $\mathcal{D}$:

$$\mathcal{L}(\mathcal{D}, \theta^{i+1}_{\mathcal{L}}) \simeq \mathcal{L}(\mathcal{D}, \theta^i) - \eta \nabla \mathcal{L}(\mathcal{D}_T, \theta^i) \nabla \mathcal{L}(\mathcal{D}, \theta^i) \tag{19}$$

Subtracting Eq. 19 from 18, we derive:

$$\mathcal{L}(\mathcal{D}, \theta^{i+1}_{\text{EWC}}) - \mathcal{L}(\mathcal{D}, \theta^{i+1}_{\mathcal{L}}) \simeq \eta \nabla \mathcal{L}(\mathcal{D}, \theta^i)(\nabla \mathcal{L}(\mathcal{D}_T, \theta^i) - \nabla \mathcal{L}_{\text{EWC}}(\mathcal{D}_T, \theta^i)) \tag{20}$$

Elastic Weight Consolidation (EWC) loss is expressed as:

$$\mathcal{L}_{\text{EWC}}(\theta^i) = \mathcal{L}(\theta^i) + \frac{\lambda}{2}\sum_i^{|\theta|} F_i(\theta^i - \theta^*)^2 \tag{21}$$

Thus, we can compute $\nabla\mathcal{L}_{\text{EWC}}(\mathcal{D}_T, \theta^i)$ as:

$$\nabla\mathcal{L}_{\text{EWC}}(\mathcal{D}_T, \theta^i) = \nabla\mathcal{L}(\mathcal{D}_T, \theta^i) + \lambda\text{diag}(F)(\theta^i - \theta^*) \tag{22}$$

Substituting Eq. 22 into Eq. 20, we obtain:

$$\mathcal{L}(\mathcal{D}, \theta_{\text{EWC}}^{i+1}) - \mathcal{L}(\mathcal{D}, \theta_{\mathcal{L}}^{i+1}) = -\eta\lambda\text{diag}(F)\nabla\mathcal{L}(\mathcal{D}, \theta^i)^T(\theta^i - \theta^*) \tag{23}$$

To determine the sign of $\eta\lambda\text{diag}(F)\nabla\mathcal{L}(\mathcal{D}, \theta^i)^T(\theta^i - \theta^*)$, we must investigate the sign of each factor. The values of $\eta$ and $\lambda$ are positive by construction. To determine the sign of $\nabla\mathcal{L}(\mathcal{D}, \theta^i)^T(\theta^i - \theta^*)$, based on the strong convexity of $\mathcal{L}$ with respect to $\theta^i$ and $\theta^*$, we have:

$$\mathcal{L}(\mathcal{D}, \theta^*) \geq \mathcal{L}(\mathcal{D}, \theta^i) + \nabla\mathcal{L}(\mathcal{D}, \theta^i)^T(\theta^* - \theta^i) + \frac{\mu_{\mathcal{L}}}{2}|\theta^i - \theta^*|^2. \tag{24}$$

Rearranging, we obtain:

$$\nabla\mathcal{L}(\mathcal{D}, \theta^i)^T(\theta^* - \theta^i) \leq \mathcal{L}(\mathcal{D}, \theta^*) - \mathcal{L}(\mathcal{D}, \theta^i) - \frac{\mu_{\mathcal{L}}}{2}\|\theta^i - \theta^*\|^2. \tag{25}$$

Since $\theta^*$ minimizes $\mathcal{L}$, the term $\mathcal{L}(\mathcal{D}, \theta^*) - \mathcal{L}(\mathcal{D}, \theta^i)$ is always negative. Moreover, $-\frac{\mu_{\mathcal{L}}}{2}\|\theta^i - \theta^*\|^2$ is also always negative, leading to:

$$\nabla\mathcal{L}(\mathcal{D}, \theta^i)^T(\theta^* - \theta^i) < 0. \tag{26}$$

Consequently, $\nabla\mathcal{L}(\mathcal{D}, \theta^i)^T(\theta^i - \theta^*)$ is positive definite.

Finally, the $\text{diag}(F)$ term is determined to be positive valued, according to the following Lemma A.1. Thus we have derived that the sign of $\eta\lambda\text{diag}(F)\nabla\mathcal{L}(\mathcal{D}, \theta^i)^T(\theta^i - \theta^*)$ is positive, which from Eq. 23 we can conclude:

$$\mathcal{L}(\mathcal{D}, \theta_{EWC}^{i+1}) - \mathcal{L}(\mathcal{D}, \theta_{\mathcal{L}}^{i+1}) < 0, \tag{27}$$

which completes the proof. $\square$

**Lemma A.1.** *Let a logistic regression model be characterized by parameters $\theta$ and trained using regularized cross-entropy loss. Then, the diagonal values of its Fisher information matrix $(diag(F))$ are strictly positive.*

### A.5 PROOF OF LEMMA A.1

*Proof.* The Fisher information matrix is the estimated value of the Hessian of the log-likelihood:

$$F = \mathbb{E}\left[\nabla^2(-\log\mathcal{L}(\theta))\right] \tag{28}$$

In logistic regression, we model the probability of a binary outcome $y$ given input $\mathbf{x}$ as:

$$P(y = 1|\mathbf{x}; \theta) = \frac{1}{1 + e^{-\theta^T\mathbf{x}}} \tag{29}$$

where $\theta$ is the vector of model parameters. For a dataset $\{(\mathbf{x}_i, y_i)\}_{i=1}^N\}$, the negative log-likelihood is:

$$-\log\mathcal{L}(\theta) = \sum_{i=1}^N\left[-y_i\log\left(\frac{1}{1 + e^{-\theta^T\mathbf{x}_i}}\right) - (1 - y_i)\log\left(1 - \frac{1}{1 + e^{-\theta^T\mathbf{x}_i}}\right)\right] \tag{30}$$

So the Hessian of the negative log-likelihood is:

$$\nabla^2(-\log \mathcal{L}(\theta)) = \begin{bmatrix} \frac{\partial^2(-\log \mathcal{L})}{\partial \theta_1^2} & \cdots & \frac{\partial^2(-\log \mathcal{L})}{\partial \theta_1 \partial \theta_d} \\ \vdots & \ddots & \vdots \\ \frac{\partial^2(-\log \mathcal{L})}{\partial \theta_d \partial \theta_1} & \cdots & \frac{\partial^2(-\log \mathcal{L})}{\partial \theta_d^2} \end{bmatrix} \tag{31}$$

As a result:

$$\nabla^2(-\log \mathcal{L}(\theta)) = \nabla^2 L(\theta) \tag{32}$$

where $d$ is the dimensionality of $\theta$. Now since the model is logistic regression and loss is regularized cross-entropy, from Eq. 7, we have:

$$\mathcal{L}(x_1) \ge \mathcal{L}(x_2) + \nabla \mathcal{L}(x_2)^T (x_1 - x_2) + \frac{\mu_{\mathcal{L}}}{2} \|x_1 - x_2\|^2, \tag{33}$$

Which is the condition of strong convexity. As a result:

$$\nabla^2 \mathcal{L} \ge \mu_{\mathcal{L}} \boldsymbol{I} \tag{34}$$

From Eq.32 and Eq. 34:

$$\nabla^2(-\log \mathcal{L}(\theta)) = \nabla^2 L(\theta) \ge \mu I \tag{35}$$

Hence:

$$\mathbb{E}\left[\nabla^2(-\log \mathcal{L}(\theta))\right] \ge \mu I \tag{36}$$

consequently:

$$\text{diag}(F) > \text{diag}(D), \quad \text{where } D_{ii} > 0, \quad \text{for all } i \tag{37}$$

which completes the proof. □

# B  IMPLEMENTATION DETAILS

All our experiments were conducted utilizing the PyTorch framework. We use the original implementation of Learning without Forgetting (LwF), Elastic Weight Consolidation (EWC), a modified version of EWC, Gradient Projection Memory (GPM), and Scaled Gradient Projection (SGP)[1]. The specifics of each algorithm's implementation are summarized in Table A1. The parameters for each algorithm such as Learning Rate (LR), Optimizer, Momentum, LR Scheduler, CL Weight, and number of Epochs are detailed.

Table A1: Table A1: Implementation Details of the Considered Algorithms for LTR benchmark.

| Algorithm | LR | Opt. | Momentum | LR Scheduler | CL Loss Weight | Epochs |
|---|---|---|---|---|---|---|
| LwF | 0.001 | SGD | 0.9 | - | 0.01 | 5 |
| EWC | 0.01 | SGD | 0.9 | - | 10 | 90 |
| Modified EWC | 0.01 | SGD | 0.9 | - | 1000 | 90 |
| GPM | 0.001 | SGD | 0 | Cosine Anneal LR | - | 100 |
| SGP | 0.001 | SGD | 0 | Cosine Anneal LR | - | 150 |

# C  DATASETS

Fig. A1 illustrates the distribution of samples among different classes and the division of the dataset into the Head and Tail sections. In the case of CIFAR100-LT with $IF = 100$, the initial partition is configured such that 5% of the samples fall within the Tail and 95% in the Head section (Classes 60 to 100 are classified as Tail). For comparison purposes, the rest of the datasets follow a similar partition threshold where 60% of the classes are assigned to the Head section.

---

[1]The code for the algorithms was obtained and modified from various open-source repositories:
https://github.com/ngailapdi/LWF
https://github.com/shivamsaboo17/Overcoming-Catastrophic-forgetting-in-Neural-Networks
https://github.com/sahagobinda/GPM
https://github.com/sahagobinda/SGP

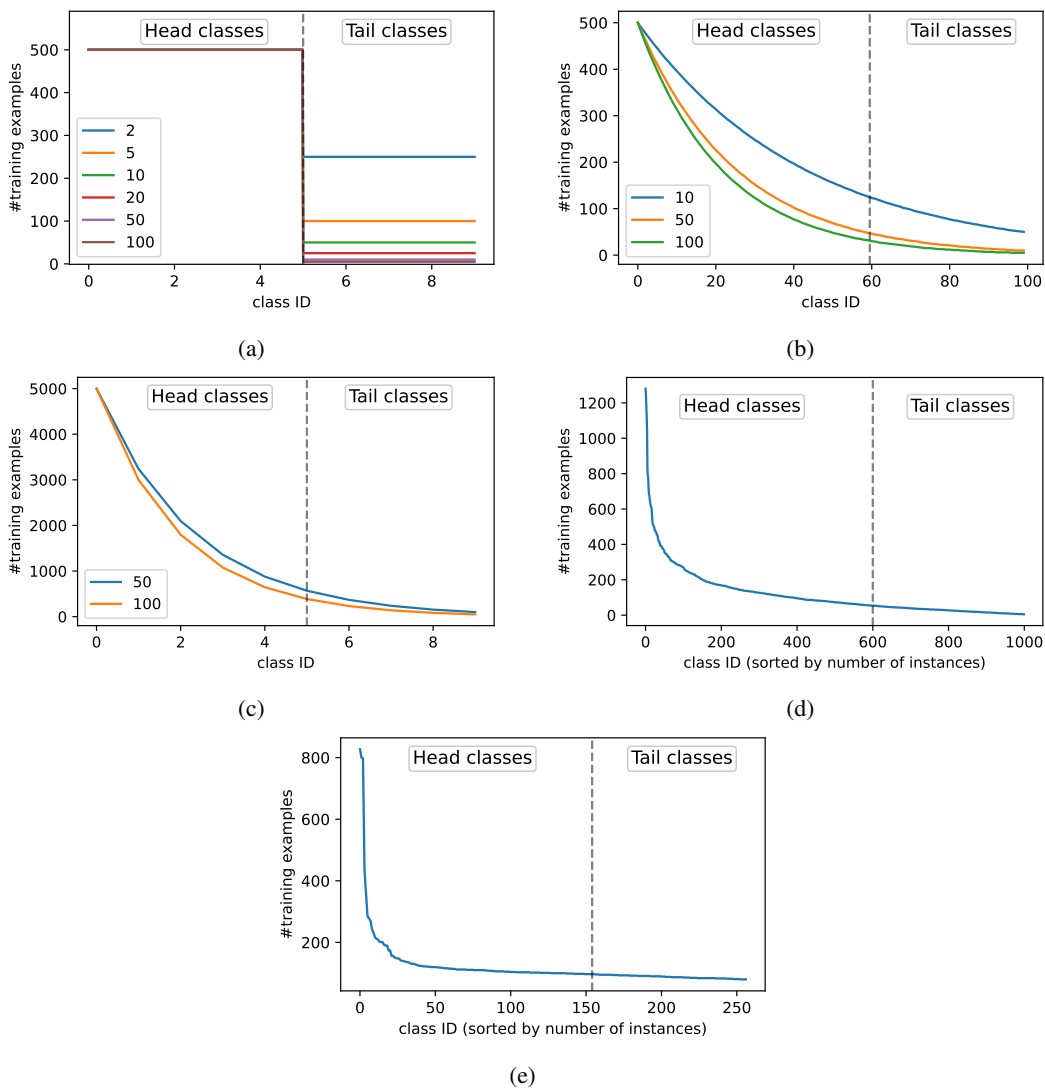

Figure A1: Class cardinality of (a) MNIST-LT, (b) CIAFR100-LT, (c) CIFAR10-LT, (d) ImageNet-LT and (e) Caltech256

# D   CL PERFORMANCE COMPARISON

Here, rather than employing the baseline for computing per-class accuracy differences, we compare the CL method, GPM (which follows the same trend but slightly worse performance than SGP), with an LTR model, WD, that exhibits similar overall accuracy. The outcomes are depicted in Fig. A2 (a). In this figure, the red bars denote classes where WD outperforms GPM, whereas the bars indicate the classes where GPM excels. We observe that GPM performs generally better on the Tail, whereas WD outperforms in Head. On average, WD's accuracy on Head classes is 4.5% higher, while GPM achieves a 9.5% higher accuracy on Tail samples. Here, we analyze the difference in per-class accuracy of GPM , Modified EWC (which exhibits similar but slightly better performance than EWC), and LwF with respect to each other, and present the results in Figs. A2 (b, c, and d). Among these three CL methods, GPM demonstrates the best results on the Tail, particularly in classes 60 to 80. LwF performs better when data is extremely limited (classes 90 to 100). The best method for Head classes is Modified EWC (outperforming GPM in 40 out of 60 Head classes), as a result of both minimizing instances of catastrophic forgetting and promoting backward transfer. These comparisons highlight that each CL method exhibits distinct behaviors when applied to the LTR problem.

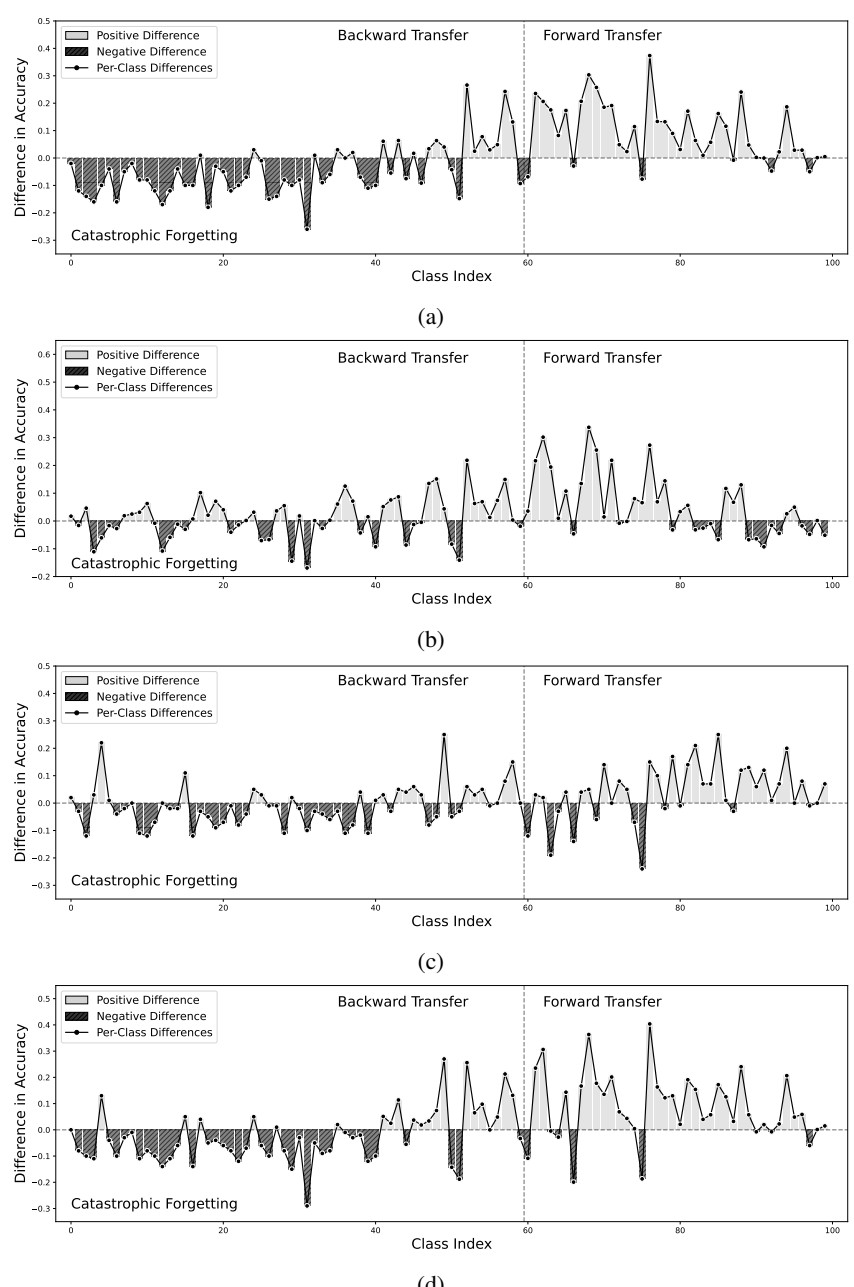

Figure A2: The difference in per-class accuracy of (a) GPM and WD, (b) GPM and LwF, (c) LwF and Modified EWC, and (d) GPM and Modified EWC.

# E    WEIGHT IMBALANCE

An interesting phenomenon observed when training models on highly imbalanced data is the presence of artificially large weights in neurons corresponding to the Head classes (Alshammari et al., 2022). The LTR solution, WD, addresses this problem by penalizing weight growth using weight decay. One way to assess the network's ability to handle LTR is by analyzing the bias in per-class weight norms. To this end, we present the per-class weight norms of the Baseline, WD, and SGP models in Fig. A3.

The figure reveals a significant imbalance in the weight norms of the Baseline model, which is naively trained on the imbalanced dataset. In contrast, the WD and SGP models exhibit more uniform weight norms across different classes. Interestingly, although SGP starts with the heavily imbalanced weights of the Baseline model, it converges towards a more uniform weight distribution without any explicit penalty on weight growth. Unlike many other CL methods that restrict the plasticity of crucial weights, GPM only constrains the direction of the weight update in the weight space, enabling the model to converge to a more balanced weight distribution. This further demonstrates the effectiveness of CL in addressing LTR.

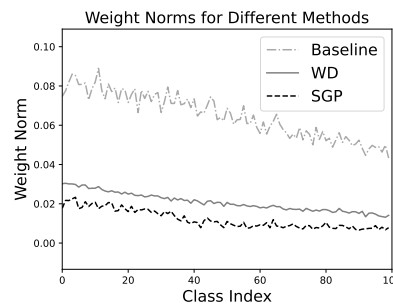

Figure A3: Per-class weight norms of the baseline, SGP, and WD.

## F    BROADER IMPACT

Dealing with imbalanced data is of paramount importance in ensuring fairness and reducing bias in AI applications, particularly in cases where the underrepresented classes correspond to minority groups. The long-tailed distribution of real-world data poses a significant challenge in achieving equitable performance for both common and rare cases. This paper's proposed algorithm, which addresses the LTR problem through the lens of CL, holds great potential in mitigating the adverse effects of class imbalance on model performance. By effectively learning from both the Head and the Tail, the proposed method can enhance the performance on underrepresented classes, leading to more fair and accurate AI models across various domains.

