# OpenReview forum: "Continual Learners are Viable Long-Tailed Recognizers"
_ICLR.cc/2024/Conference — ICLR 2024 Conference Withdrawn Submission_

### Official Review · Reviewer_U9N3 · 2023-10-18

**Soundness:** 2 fair
**Presentation:** 3 good
**Contribution:** 3 good
**Rating:** 3
**Confidence:** 3

**Summary:**

This article innovatively introduces the concept of continual learning into the long-tail learning process. The article combines theory and experiments to demonstrate the parameters bound between trained on head classes and the entire dataset. Experimental results also show that continual learning methods can effectively improve model performance in long-tail distributions.

**Strengths:**

1. The article is highly readable, with well-explained theory and reasoning, and comprehensive appendices. It is easily understandable.
2. The article presents a novel viewpoint, creatively linking two different domain works and integrating them organically.
3. Although the experimental results fall slightly short of the state-of-the-art (SOTA), they are still acceptable.

**Weaknesses:**

1. The article focuses on theoretical findings, but the proven relationship between $\theta^*$ and $\theta_H^*$ is not the most crucial aspect of interest. The omission of discussing $\theta_{HT}^*$ in relation to $\theta^*$ and $\theta_H^*$ is problematic point.
2. Similarly, the argument that CL techniques are superior to naively retraining (i.e., fine-tuning) is not crucial. In current methods, fine-tuning already involves weighted adjustments, even exclusively on the head. Equation 17 oversimplifies the problem.
3. Continual learning typically involves multiple rounds of weight updates, but in this article, it is only divided into two categories: head and tail classes. Would better performance be achieved by designing more categories or exploring techniques that require only an additional training round?
4. The article lacks an ablation study. For example, how does the proportion of divisions affect the results? What performance can be achieved with more categories, as suggested in W3? These aspects are worth exploring.

**Questions:**

See Weakness.

In fact, I really appreciate the insights of this article. At first glance, it felt refreshing and intriguing. However, upon closer reading, I feel that there may be some gaps between what can be proven and what is actually needed in the long-tailed domain. The proofs do not address the most critical issues, hence the given rating. I am unsure if this rating is overly strict, and I will also consider the opinions of other reviewers to adjust my score. Overall, I believe this perspective is worth exploring, but when theory cannot cover everything, conducting more experiments or proposing a new design approach may be a better endeavor.

---

### Official Review · Reviewer_FdyD · 2023-10-27

**Soundness:** 3 good
**Presentation:** 3 good
**Contribution:** 2 fair
**Rating:** 5
**Confidence:** 4

**Summary:**

This paper introduces a theorem setting an upper bound on weight discrepancies when training on imbalanced dataset partitions. Leveraging this theorem, the authors provide a novel approach for using CL in the LTR problem, verified by experiments showing performance improvements.

**Strengths:**

1. Theoretical Foundation: The authors introduce a novel theorem that establishes an upper bound on the difference in weights when a model is trained under the LTR setting;

2. Application to the LTR Problem: Based on the aforementioned theorem, the paper leverages CL solutions for the LTR problem. The authors provide proof for the effectiveness of CL when it comes to LTR setting, i.e., the Head and Tail set in a sequential manner;

3. Empirical Validation: The theoretical propositions and methodologies are supported by extensive experimental results.

**Weaknesses:**

The weaknesses of this paper include:

In the related works section, the authors do not sufficiently categorize continual learning into "expansion-based, regularization-based, and memory-based approaches". I recommend that the authors check some surveys on continual learning, such as [1], to gain clarity. While the main focus is LTR, it would be beneficial for the authors to identify two crucial assumptions in the related work: whether the proposed method requires memory and whether the proposed method needs a task ID during inference.

The paper's novelty is ambiguous. Several prior works have already explored the benefits of using CL for the LTR problem. It would be beneficial for the authors to reference these works and elucidate their unique contributions.

Additionally, the authors have overlooked a vital baseline in CL, specifically the "balanced CL", as cited in references [2-5]. It's essential for them to consider and differentiate their work from these foundational studies. Since it is highly related to the LTR settings.

[1] A Comprehensive Survey of Continual Learning: Theory, Method and Application

[2] Layerwise optimization by gradient de- composition for continual learning.

[3] Online continual learning from imbalanced data.

[4] TKIL: Tangent Kernel Optimization for Class Balanced Incremental Learning

[5] Continual learning in low-rank orthogonal sub- spaces.

**Questions:**

Please refer the weakness 1. The author should identify this method belongs to which CL method categories and then select the proper baseline for comparisons.

---

### Official Review · Reviewer_PwBM · 2023-10-31

**Soundness:** 3 good
**Presentation:** 3 good
**Contribution:** 3 good
**Rating:** 5
**Confidence:** 4

**Summary:**

The paper proposes to use continual learning (CL) techniques to perform long-tailed recognition (LTR). The paper shows a series of theorems that, under the assumption of strong convexity, show that the weights of a learner trained on a long-tailed dataset are within some neighborhood of the network trained only on the more common (head) classes. This is used to motivate a two-step approach to long-tailed recognition, first train on the common classes and then on the less-common (long-tail) classes. The proposed approach is evaluated and obtains good results (not state-of-the-art).

**Strengths:**

- the idea to use continual learning techniques for long-tailed recognition is new I think
- the authors do a good job in providing a principled approach of using CL for LTR (including the explanation in Figure 1, and the theorems for the case of strong convexity).
- Results are quite competitive on several LTR benchmarks

**Weaknesses:**

- Continual learning typically works with the assumption that there is no or limited access to previous task data. However, in the discussed scenario this is not the case, and the authors could just put the whole first task (head set) in the buffer. Typically, this is considered an upper bound in continual learning. For me, it is not clear why not to exploit this possibility. And within that setup, it is not so clear what anti-fogetting mechanisms would be needed.

- I would have preferred a bit more analysis where the authors zoom in on particular methods and change the amount of anti-forgetting (continual learning techniques) that are applied. For example, for EWC one can change the lambda parameter and more from FT (full plasticity) to a very high lambda (max. stability); showing the accuracy as a function of the lambda would be very insightful. And for me, more convincing that CL methods could be a tool to improve LTR. (the same holds for other methods, like LwF).

- In general, backward transfer is not common in CL. I am somewhat surprised about the results in Figure 4. Normally, backward and forward transfer are defined for task-aware incremental learning (TIL). I understand here that the results are class-incremental (CIL). How are forward and backward transfer computed ?

**Questions:**

I would like the authors to answer the weaknesses. Especially, the discussion on why not replaying all previous data -- which from a CL point of view is an upperbound -- needs to be discussed.

I appreciate the authors work for a principled introduction of CL in LTR, however, fundamentally, I am not convinced it is really useful, since you have the possibility of full dataset replay. I also think that the analysis could be improved and needs some more clarifications.

minor:
I prefer the methods names, rather than author names in table  1 (but maybe this is normal in LTR literature? )

---

### Official Review · Reviewer_bvQv · 2023-11-01

**Soundness:** 4 excellent
**Presentation:** 3 good
**Contribution:** 3 good
**Rating:** 5
**Confidence:** 4

**Summary:**

This work introduces a novel approach to address Long-Tailed Recognition (LTR) by using Continual Learning (CL) on imbalanced datasets. The authors present theoretical theorems demonstrating that CL can effectively optimize models to perform well on both majority and minority class subsets in LTR. They validate their approach using various datasets, including standard LTR benchmarks and a real-world dataset (Caltech256), showing that CL methods outperform baseline models and tailored LTR approaches, suggesting the potential of leveraging CL techniques to address the LTR challenge more effectively.

**Strengths:**

- The integration of the concept of continual learning into the long-tailed recognition problem is indeed intriguing and represents a novel approach.
- The authors have provided a theoretical proof that demonstrates the proximity of the weights learned from a long-tailed dataset to those exclusively trained on the head dataset.
- A comprehensive set of experiments has been conducted to illustrate the effectiveness of continual learning in addressing the long-tailed recognition problem.

**Weaknesses:**

- The performance of continual learning in long-tailed settings is not particularly noteworthy, even in the absence of recent state-of-the-art long-tailed recognition methods like RIDE, ACE, SSD, and PaCo in the comparative evaluation.
- The inclusion of additional results derived from the iNaturalist-LT dataset would be beneficial for a more comprehensive assessment of the effectiveness of the proposed method.
- The theorems and statements presented in Section 3.4 provide a level of assurance that Continual Learning (CL) surpasses the performance of the baseline cross-entropy loss. However, it remains open to question whether CL offers a distinct advantage over conventional long-tailed recognition methods.

**Questions:**

- Because the cardinality distribution is not particularly discrete, how do you separate the head and tail? What criteria could be used?
- How about learning tail datasets first, followed by head datasets?